# Ontogenetic Changes in the Chemical Profiles of *Piper* Species

**DOI:** 10.3390/plants10061085

**Published:** 2021-05-28

**Authors:** Anderson Melo Gaia, Lydia Fumiko Yamaguchi, Camilo Guerrero-Perilla, Massuo Jorge Kato

**Affiliations:** Institute of Chemistry, University of São Paulo, Av. Prof. Lineu Prestes, 748, São Paulo 05508-900, SP, Brazil; andersongaia@gmail.com (A.M.G.); lydyama@iq.usp.br (L.F.Y.); camilogp@iq.usp.br (C.G.-P.)

**Keywords:** ontogeny, *Piper*, secondary metabolites, seedlings

## Abstract

The chemical composition of seedlings and adult plants of several *Piper* species were analyzed by ^1^H NMR spectroscopy combined with principal component analysis (PCA) and HPLC-DAD, HPLC-HRESIMS and GC-MS data. The chromatographic profile of crude extracts from leaves of *Piper* species showed remarkable differences between seedlings and adult plants. Adult leaves of *P. regnellii* accumulate dihydrobenzofuran neolignans, *P. solmsianum* contain tetrahydrofuran lignans, and prenylated benzoic acids are found in adult leaves of *P. hemmendorffii* and *P. caldense*. Seedlings produced an entirely different collection of compounds. *Piper gaudichaudianum* and *P. solmsianum* seedlings contain the phenylpropanoid dillapiole. *Piper* *regnellii* and *P. hemmendorffii* produce another phenylpropanoid, apiol, while isoasarone is found in *P. caldense*. *Piper richadiaefolium* and *P. permucronatum* contain dibenzylbutyrolactones lignans or flavonoids in adult leaves. Seedlings of *P. richardiaefolium* produce multiple amides, while *P. permucronatum* seedlings contain a new long chain ester. *Piper* *tuberculatum, P. reticulatum* and *P. amalago* produce amides, and their chemistry changes less during ontogeny. The chemical variation we documented opens questions about changes in herbivore pressure across ontogeny.

## 1. Introduction

The genus *Piper* belongs to the family Piperaceae and it is considered a megadiverse group among Angiospermae [1]. *Piper* is the second largest genus in the Piperaceae with more than 1000 species distributed in both hemispheres of the Tropics [2,3], and it can be found in dense populations in tropical forests. Members of the genus *Piper* have specific interactions with pollinators, seed dispersers, and herbivores [4,5,6]. *Piper* is chemically very diversified and the most common class of secondary compounds found are amides. Characteristic amides include piperine and related compounds isolated from *P. nigrum* [7]. While some species are rich in amides such as *P. tuberculatum* [8], *P. scutifolium* and *P. hoffmanseggianum* [9], other characteristic compounds found in *Piper* are lignans from *P. solmsianum* [10], neolignans from *P. regnellii* [11] and *P. decurrens* [12], and acid benzoic derivatives in *P. aduncum, P. crassinervium* [13,14,15,16], *P. hispidum* [17] and *P. gaudichaudianum* [15,18]. Multiple compounds isolated from *Piper* species have antifungal, antimicrobial and insecticidal activities [2,9,19,20,21,22]. 

Phytochemical studies on the isolation of bioactive compounds, or descriptions of the role of secondary metabolites in mediating plant-insect interactions, usually focus on adult plants. Ethnobotanical and bioprospecting studies also often target adult plants. Data on chemical variation in plants come from studies on responses to abiotic and biotic factors including light, nutrients, herbivory, and pathogens, but some important work has also documented changes in secondary metabolites during ontogeny. Chemical profiles are known to change during the ontogeny of several *Piper* species, suggesting different biosynthetic pathways become active during development. *Piper gaudichaudianum* seedlings, for example, produce the phenylpropanoids apiol and dillapiole, while the prenylated benzoic acid, gaudichaudianic acid, is found in adult leaves [23]. 

Herbivory is most damaging to shrubs, such as *Piper*, early in ontogeny [24], and the plant-age hypothesis predicts young plants should be well defended because they are particularly vulnerable to herbivory [24]. Research supports the hypothesis that juvenile shrubs may produce more or different chemical defenses than mature individuals [23,25]. Herbivory is one of the most severe threats at the early stage of shrubs, such as in *Piper* species [26]. Ontogenetic changes in defenses are common [27,28], as herbivore host searching is limited and survival is higher among seedlings which are chemically distinct from their mother plants [29,30]. In addition, across species, herbivory is particularly limiting in dense patches of conspecific seedlings [31,32,33,34]. Plant chemotypes are more variable than phylogeny would suggest [35,36], further indicating that variation in plant chemistry enables survival by limiting herbivory. Together, these patterns suggest that studying plant secondary metabolites early in ontogeny may lead to a better understanding of phytochemical diversity. In the present work, we document ontogenetic changes in secondary metabolites in nine *Piper* species, showing that changes in phytochemistry are actually quite common during development.

## 2. Results

### 2.1. Phenylpropanoids as a Common Feature in Seedlings of Piper Species

Previous work on ontogenetic variation of the chemistry of *P. gaudichaudianum* shows that the main chemical constituents of seedlings are the phenylpropanoids apiol (**1**) and dillapiole (**2**). During plant development, these compounds are replaced by gaudichaudianic acid (**6**) a prenylated benzoic acid [23]. ^1^H NMR data comparing the profile of adult and seedling leaves (3 months old) of *P. regnellii* (Miq.) C. DC., *P. solmsianum* C. DC., *P. caldense* C. DC. and *P. hemmendorffii* C. DC., indicated that seedlings produce phenylpropanoids similar to *P. gaudichaudianum*. A PCA plot indicates clear dissimilarity between seedling and adult leaves. The scores obtained by the combination of PC1 and PC2 explained 61% of the total variance (Figure 1A) and showed a clear differentiation between seedling and adult leaves. Seedlings of *P*. *gaudichaudianum* and *P. solmsianum* both produced apiol, and *P. hemmendorffii* and *P. regnellii* seedlings both produced apiol and dillapiole. The loading plot (Figure 1B) shows the main signals in the ^1^H NMR spectrum responsible for separating the samples. Methoxy groups (δ 3.82 and 3.86) of grandisin (**12**) were detected in adult leaves of *P. solmsianum*. Seedling leaves differed and were characterized by the methylenedioxy (δ 5.94) and methoxy (δ 3.84) groups of the phenylpropanoids apiol (**1**) and dillapiole (**2**) (Figure 1C).

Detailed chemical profiles were obtained using HPLC-DAD data (Figure 2) of ethyl acetate extracts from adult leaves of *P. gaudichaudianum*, *P. regnellii* and *P. solmsianum.* Gaudichaudianic acid (**6**) was the major constituent of *P. gaudichaudianum*; conocarpan (**14**) was found in *P. regnellii*, and grandisin (**12**) was prominent in *P. solmsianum*. The chemical profile of *P. solmsianum* (Appendix A) and *P. gaudichaudianum* (Appendix A) seedlings was characterized by the phenylpropanoid apiol (**1**) in *P. solmsianum*. Dillapiole (**2**) was essentially the sole compound in *P. regnellii* (Appendix A) and *P. hemmendorffii* (Appendix A) seedlings, respectively). 

The phenylpropanoid isoasarone (**4**) was by far the main compound in seedlings of *P. caldense* (Figure 2; Appendix A). The main compound in adult leaves of *P. hemmendorffii* was the benzoic acid **9** [37], and the primary compound in adult *P. caldense* was caldensinic acid (**11**) (Appendix A) [38]. Despite production of benzoic acid derivatives and lignoids in adult stages, all these species produce phenylpropanoids as the major compounds early in ontogeny. *P. hemmendorffii* and *P. caldense* produce dillapiole and isoasarone early in ontogeny, and, as adults, *P. hemmendorffii* produces the benzoic acid **9,** while *P. caldense* produces the benzoic acid **11.** *Piper regnellii* produces dillapiole (**2**) at the seedling stage and benzofuranoid neolignans (**14**, **15a**, **15b**) as adults. *P. solmsianum* also produces dillapiole as seedlings, but adults contain tetrahydrofuran lignan (**12**). In summary, these four species, defined as Group I, all produce phenylpropanoid as seedlings. Apiol and dillapiole are dominant compounds and the methylenedioxy function present in their structures is associated with relevant biological properties such as synergisms with insecticides [39]. Interestingly, the isoasarone (**4**) only has methoxy groups but, it is predicted to have antimicrobial activity in extracts from rhizomes of *Acorus calamus* [40]. 

### 2.2. Changes in the Secondary Metabolite Profiles during Ontogeny

The two extremes in composition from phenylpropanoid at the age of 3 months to the compounds observed at adult stages was further expanded to determine how this shifting in chemical composition is taking place throughout development. In the previous case of *P. gaudichaudianum*, the transition between the two profiles allowed the detection of methyl taboganate (**7**) and a chromone (**8**), two compounds biosynthetically related to gaudichaudianic acid (**6**). The detailed study of additional *Piper* species during ontogeny over the timeframe of 3–15 months would provide data on the transition between extremes of composition from phenylpropanoids to lignoids in *P. solmsianum, P. regnellii*, and to prenylated benzoic acid in *P. caldense*. 

#### 2.2.1. *Piper solmsianum*

The analysis of the chemical profile of seedling leaves at 3, 6, 9, 12 and 15 months and adult leaves of *P. solmsianum* was carried out with ^1^H NMR (500 MHz; Appendix A) and these data were subjected to principal component analysis. PC1 and PC2 explained 89% of data variance, and the score plot (Figure 4) indicated the two ontogenetic stages differ chemically. The loading plot (Appendix A) revealed that the main variables responsible for separating the groups were the chemical shifts at δ 3.84 of the methoxy group of apiol (**1**) and the signals at δ 5.94 and 5.92 (methylenedioxy) and δ 3.30 and 3.28 (methylene of the allylic group) assigned to apiol (**1**) and myristicin (**5**), respectively (Appendix A). For adult leaves, the chemical shifts corresponding to the grandisin (**12**) were observed at δ 6.62 and were attributed to aromatic hydrogens and at δ 3.82 and 3.88 to the methoxy groups [41].

The HPLC-UV chromatogram (Figure 5) of the adult *P. solmsianum* leaves showed the phenylpropanoid elemicin (**3**) and the tetrahydrofuran lignan grandisin (**12**) were the main constituents. The analysis of the crude extracts of seedlings at 3 and 6 months showed apiol (**1**) was a main constituent. Profiles of leaf extracts at 9 and 12 months also showed **1** as a main constituent, but at these stages, myristicin (**5**) and elemicin (**3**) were also detected. At 15 months, chemical diversity was greatest because the chemical profiles were shifting between seedling and adult stages. All compound identities were confirmed either by GC-MS or, HPLC-HRESIMS analysis and comparison with standards. 

#### 2.2.2. *Piper regnellii*

A PCA analysis was performed using ^1^H NMR spectra (500 MHz) data of adult and seedling leaves of *P. regnellii* (Appendix A). PC1 and PC2 accounted for 80% of data variance, and the score plot (Figure 6) indicated a clear distinction between seedling leaves (3–12 months) and adult leaves, while the leaves at the intermediate stage of 15 months appeared between these extremes. The loading plot (Appendix A) showed that the main variables separating the different ontogenetic states was the presence of dillapiole (**2**) in seedlings (Appendix A), with NMR data at δ 3.74 and 4.00, assigned to the methoxy groups, at δ 5.88 to the methylenedioxy group, and at δ 3.30 assigned to the allylic methylene. In the adult leaves, the dihydrobenzofuran neolignan conocarpan (**14**) was identified by the chemical shifts at δ 1.84 corresponding to the methyl group, and at δ 6.82 and 7.12 to its aromatic hydrogens [11].

The analysis carried out by HPLC-UV (Figure 7) for the adult leaves of *P. regnellii* confirmed the main constituents as conocarpan (**14**), eupomatenoid-6 (**15a**) and eupomatenoid-5 (**15b**). Dillapiole (**2**) was identified as a major component in the crude extract of seedling leaves at 3 and 6 months old, although extracts from 9 month old leaves revealed that the neolignans gradually appeared at 12 and 15 months old, and the neolignans (**14**, **15a** and **15b**) became dominant while apiol and dillapiole were not detectable. 

#### 2.2.3. *Piper caldense*

Ontogenetic analysis of *P. caldense* with ^1^H NMR of the crude extracts combined with PCA (Figure 8) showed a clear dissimilarity between seedling and adult leaves. PC1 and PC2 explained 79% of the variance. The loading plot (Appendix A) showed that the main variables associated with the separation of the seedling leaves, corresponded to the chemical shifts of the phenylpropanoid isoasarone (**4**). The chemical shifts observed at δ 3.80, 3.82 and 3.88 were assigned to the methoxy groups, and the doublet at δ 3.32, associated with the signals at δ 6.52 and 6.68, corresponded with the allylic methylene and aromatic hydrogens of isoasarone (Appendix A). For the adult leaves, chemical shifts of caldensinic acid (**11**) were observed at δ 1.56, 1.62, 1.64 and 1.72 and were assigned to the methyl groups of prenyl units, and the signals of methylene observed at δ 2.02, 2.04, 2.12, 2.22, 3.36 (benzylic), and 5.10 ppm were associated with the four prenyl units (Appendix A) [38]. 

The PCA plots were based on HPLC-UV analysis (Figure 9). As expected, the adult leaves contained caldensinic acid (**11**) as the main constituent, which was previously isolated and was used as a standard [38]. A major constituent in 3 months old leaves was isoasarone (**4**) [42]. Isoasarone was present until seedlings were 12 months old, and then was replaced by **11** [38]. It is interesting that caldensinic acid, but not isoasarone, is a major constituent of seeds but does not seem to be translocated to seedling leaves. The young leaves of the adult plants also contain **11,** and other minor compounds that were not characterized. 

### 2.3. Piper Species Producing Amides

The group of species consisting of *P. tuberculatum* Jacq.*, P. amalago* L. and *P. reticulatum* L., is characterized by the production of amides in their adult leaves [8,43,44,45]. The PCA of ^1^H NMR data of crude extracts from seedlings and adult leaves shows *P. amalago* and *P. tuberculatum* are chemically distinct, but their chemistry does not change across ontogeny (Figure 10). *Piper reticulatum* seedlings and adults are chemically distinct. Piplartine (**17**) is the dominant compound in adult and seedling leaves of *P. tuberculatum* (Appendix A)*,* as evidenced in the loading plot by signals assigned to methoxy groups (δ 3.86 and 3.88) of piplartine [8]. *Piper amalago* (Appendix A) had signals assigned to nigrinodine (**21**) for both adult and seedling leaves (δ 5.90 and 5.92 of methylenedioxy groups) (Appendix A) [44]. 

Amides were the dominant compounds in adult and seedling leaf extracts of *P. tuberculatum, P. amalago* and *P. reticulatum.* Piplartine (**17**) was the major metabolite in seedlings and adults of *P. tuberculatum*, and nigrinodine (**21**) was the primary component of *P. amalago* adults and seedlings. (3*E*,5*E*,14*E*)-*N*-pyrrolidyleicosa-3,5,14-trienamide (**22**) is predominant in *P. reticulatum* adults, and seedling leaves contained the dihydrowisanidine (**20**) as the major compound (Appendix A). HPLC-UV data were consistent with results obtained by PCA and revealed that the changes in *P. reticulatum* are more significant between the two stages, but with a variable relative content of **20** and **21** (Figure 11). The GC-EIMS analyses of the crude extracts showed *P. tuberculatum* seedlings and adults had piplartine (**17**); *P. amalago* seedlings and adults produced nigrinodine (**21**), and *P. reticulatum* seedlings and adults contained dihydrowisanidine (**20**). Piplartine was identified using a standard, but nigrinodine and dihydrowisanidine were characterized by GC-EIMS data.

### 2.4. P. permucronatum Yunck. and P. richardiaefolium Kunth

Seedlings of *P. permucronatum* and *P. richardiaefolium* grouped together in the PCA score and loading plots based on ^1^H NMR data, but the adult samples were chemically distinct (Figure 12; Appendix A). PC1 and PC2 explained 76% of data variance. The chemical shifts observed for these samples were assigned to the amides piplartine (**17**) and piperine (**18**) (Appendix A) and thus, ^1^H NMR data of *P. tuberculatum* were included in the analysis. The chemical shifts at δ 5.98 corresponded to methylenedioxy hydrogens, and at δ 6.60 and 6.72 for aromatic hydrogens of piperine (**17**). The presence of piplartine was determined by the signals at δ 3.86 and 3.88 corresponding to methoxy hydrogens, and at δ 6.80 for aromatic hydrogens (Appendix A) [43,46]. Identifications were confirmed with chromatographic analyzes using standard compounds.

The ^1^H NMR variables that influenced the separation of the adult leaves of *P. permucronatum* (Figure 11; Appendix A) were attributed to the chemical shifts of the flavanone sakuranetin (**16**), observed at δ 6.04, 6.06, 6.88 and 7.32 ppm, corresponding to the aromatic hydrogens, and in addition to the methoxy group at δ 3.80 [47,48]. The identity of sakuranetin was confirmed with ^1^H NMR and GC-MS analyses of a semi-purified fraction. 

The analysis of the crude extracts from adult leaves of *P. richardiaefolium* by HPLC-UV, GCMS and ^1^H NMR indicated the presence of two dibenzylbutyrolactone and two dibenzylbutyrolactol lignans. The lignans were identified as hinokinin (**13a**), kusunokinin (**13b**), cubebin (**13c**), and **13d** [44] as previously reported for *P. richardiaefolium* [44,49,50]. Nevertheless, its seedling leaves (Appendix A) contained piperine (**19**) [51], piplartine (**17**) and pellitorine (**18**) [52], with these identities supported by ^1^H NMR (Appendix A), HRESIMS data and the use of standard compounds. 

The main constituent of the adult leaves of *P. permucronatum* was the flavanone sakuranetin (**16**; Appendix A). HPLC-UV (Figure 13) and GC-EIMS analyses of the crude extracts from adult leaves confirmed its identification by comparison with reported data [44,53]. HPLC-UV data of the seedling leaves showed two major metabolites not detected in the adult leaves.

Therefore, the content of crude extracts from seedling leaves of *P. permucronatum* was further investigated with ^1^H NMR (Appendix A), GC-EIMS, and HPLC-HRESIMS data. One of the compounds was identified as the isobutyl amide pellitorine (**18**) based on the protonated quasimolecular ion [M+H]^+^ at *m/z* 224.2032 (calculated *m/z* 224.2014) observed in its HRESI spectrum which is compatible with the expected molecular formula of C_14_H_25_NO [52]. The second compound (**23**) had a molecular formula assigned as C_15_H_16_O_4_, based on the positive mode HRESIMS (Appendix A). Its spectrum displayed a quasimolecular ion peak [M+H]^+^ at *m/z* 261.1111 Da (calculated *m/z* 261.1112) and a sodium adduct ion [M+Na]^+^ at *m/z* 283.0937 (calculated *m/z* 283.0946). Since these data did not allow its structural determination, further purification was required from the crude extract of seedlings. The ^1^H NMR spectrum of purified compound **23** (Appendix A) showed two intense singlets at δ 3.73 (3H) and at 5.93 ppm, assigned to a methoxy and a methylenedioxy group, respectively. Signals corresponding to three aromatic hydrogens were observed as two doublets at δ 6.73 (8.0 Hz) and at δ 6.66 (1.5 Hz) and a doublet of doublets at δ 6.61 (8.0 and 1.5 Hz) compatible with a 1,3,4-trissubstituted aromatic ring. The 3,4-methylenedioxybenzyl moiety was determined based on a peak at *m/z* 135 Da, typical of a methylenedioxyphenyl tropylium cation in its EIMS (Appendix A). The ^13^C NMR data further confirmed this moiety and revealed two signals corresponding to the methylene groups at δ 34.8 and 35.0 ppm. The methyl ester carbonyl was assigned based on signals at δ 167.2 (CO) and 51.5 (OMe). The ^1^H NMR data also displayed a doublet at δ 5.79 (15.0 Hz) and a multiplet at δ 6.09 corresponding to the α,β-conjugated double bond. The observed HMBC correlations (Appendix A) indicated long distance coupling between the hydrogens H-5 (δ 6.73) and C-1 (δ 134.9), between H-7 (δ 2.67) and C-6 (δ 121.2), between H-2 (δ 6.66) and C-7 (δ 34.8), and between H-9 and H-10 (δ 6.14) and C-7 (δ 34.8) and C-8 (δ 35.0) (Appendix A). Therefore, the structure for the novel compound **23** was determined as methyl (2*E*,4*E*)-7-(Benzo[*d*][1,3]dioxol-5-yl)hepta-2,4-dienoate.

## 3. Discussion

The phytochemical analyses of *Piper* species across ontogeny allowed the characterization of three main groups of species according to the type of secondary metabolites found in their leaves. The Group I with *P. solmsianum, P. regnellii, P. caldense* and *P. hemmendorffii* showed seedlings with phenylpropanoids like *P. gaudichaudianum.* Seedlings of *P. solmsianum* contained the phenylpropanoid apiol (**1**); *P. regnellii* and *P. hemmendorffii* seedlings contained dillapiole (**2**); isoasarone (**4**) was produced in *P. caldense* seedlings and *P. gaudichaudianum* produced apiol (Figure 14). This phenylpropanoid-based metabolism in seedlings changes during development until seedlings reach 12-15 months and begin producing prenylated benzoic acids (*P. gaudichaudianum, P. caldense* and *P. hemmendorffii*), tetrahydrofuran lignans (*P. solmsianum*), or benzofuranoid neolignans (*P. regnellii*). 

Group II of *Piper* (*P. tuberculatum, P. amalago* and *P. reticulatum*) is characterized by amides in seedling and adult leaves (Figure 14). No major ontogenetic changes were detected in *P. tuberculatum* and *P. amalago.* Despite the separation of *P. reticulatum* seedling and adult stages in the PCA of ^1^H NMR data, they were similar overall. Discrimination was caused by a broad, unidentified peak. Finally, Group III, consisting of *P. richardiaefolium* and *P. permucronatum,* was characterized by dibenzylbutanoid lignans (**13a**–**13d**) in *P. richardiaefolium* and sakuranetin (**16**) in *P. permucronatum* adults. Seedlings contained amides such as pellitorine (**18**) in both species. In addition, while *P. richardiaefolium* contained piplartine (**17**) instead of lignans, for *P. permucronatum* the major component (**23**) was elucidated as the methyl ester of the same carboxylic acid of nigrinodine (**21**), which was not detected in *P. permucronatum* (Figure 14). 

Overall, these data showed gradual changes in the metabolic profile of Group I during ontogeny. Seedlings in Group I contain various phenylpropanoids. Changes in the chemical composition of these species during ontogeny may provide better fitness to cope with pressure from insect herbivores and microorganisms at early stages of plant development. Phenylpropanoids and amides are effective defenses against herbivores and pathogens [54,55,56,57,58,59,60,61]. Seedlings that are chemically distinct from their parent plants survive better [30], demonstrating the importance of ontogenetic variation in chemical defenses. The phenylpropanoids detected in seedlings are recognized as insecticidal and antifungal compounds [55,56,57], and the presence of methylenedioxyphenyl groups is particularly interesting since methylenedioxyphenyls can interact synergistically with pyrethroids [39,58]. It is interesting that species in Group II, which produces amides, do not change chemically during ontogeny; perhaps this is due to the high toxicity of amides. Amides have insecticidal and repellent effects [59,60,61]. In sum, studies of seedling chemistry can contribute to a better understanding of the evolution of plant defenses and insight into biosynthetic pathways.

## 4. Materials and Methods 

### 4.1. Plant Material

Seeds and leaves of adult plants (*P. gaudichaudianum, P. regnellii, P. solmsianum, P. caldense*, *P. hemmendorffii,* and *P. amalago*) were collected from the garden of the Institute of Chemistry (University of São Paulo, Brazil). The species *P. richardiaefolium* and *P. permucronatum* were collected in the Parque Nacional de Itatiaia; the species *P. tuberculatum* was collected at the Institute of Chemistry (UNESP, Araraquara), and *P. reticulatum* in the Floresta Nacional do Tapajós (Santarém, PA) (Permit Sisbio #15780). The species were identified by Dr. Elsie F. Guimarães from the herbarium of Fundação Jardim Botânico do Rio de Janeiro and Eric J. Tepe from the University of Cincinnati, and the voucher specimens were deposited in the herbaria of the Fundação Jardim Botânico do Rio de Janeiro and at the University of Cincinnati. Seeds were removed from mature *Piper* spikes, soaked in water and sterilized with bleach (3%) for 3 min and rinsed with distilled water. The dry seeds were spread over a mixture of commercial soil and sand for germination under controlled conditions at 25 ± 2 °C and with a 16 h photoperiod. Typically, the germination took 2–3 weeks, and the resulting seedlings were harvested at 3, 6, 9, 12, and 15 months. 

### 4.2. Extraction and HPLC-DAD Analysis

Fresh plant material of adult plants (third leaves in a branch) and seedlings (pool of three plants) was frozen in liquid nitrogen and freeze-dried. The ground plant material was extracted three times with ethyl acetate overnight at room temperature. The crude extracts obtained were concentrated to dryness and residual water was eliminated using a vacuum concentrator (CentriVap, Labconco, Kansas City, MO, USA).

Dried crude extracts were dissolved in appropriate amounts of methanol to concentrations of 1.0 mg/mL and filtered through 0.45 µm PTFE filters. Aliquots of 20 µL were analyzed by HPLC-DAD and carried out with an HPLC Shimadzu with a binary pump system (LC-10 AD) equipped with a diode array detector (SPD-10 AVP), Phenomenex Luna column (C18, 5 μm, 250 × 4.6 mm) and an SCL-10 control unit. Data were collected with Class-VP software and the mobile phase consisted of MeOH:H_2_O (1% AcOH), the flow rate was 1.0 mL/min, and the oven temperature was set at 40 °C. The chromatographic conditions used for *P. gaudichaudianum, P. regnellii, P. solmsianum*, *P. hemmendorffii* and *P. caldense* started with 60% MeOH (4 min) and increased to 100% MeOH at 30 min, and conditions were maintained for 5 min with a flow rate of 1 mL/min. The crude extracts from *P. tuberculatum*, *P. amalago*, *P. reticulatum, P. permucronatum* and *P. richardiaefolium*, were analyzed starting with 50% MeOH; the remaining gradient was as described above. The identification of the compounds included in the crude extracts was identified based on the retention times and UV spectra and chromatographic comparison of compounds with standards (**1**–**5**, **6**, **10**–**14**, **16**, **17**, **19**). 

### 4.3. HRESIMS

High-resolution mass spectrometric analyses with electrospray ionization (ESI) were performed using a Bruker micrOTOF–QII, mass spectrometer. The ESI interface was operated in a positive mode with 4.5 kV in the capillary and 0.5 kV in the end plate offset. The pressure of the nebulization gas was 0.4 Bar; the drying gas was maintained at a flow rate of 8 L/min at 200 °C. The collision and the quadrupole energy were set to 12 and 6 eV, respectively. RF1 and RF2 funnels were programmed to 150 and 200 Vpp, respectively. The mass spectra were calibrated using sodium formate.

### 4.4. GC-MS Analysis 

Samples were analyzed with a Shimadzu GC-2010 Plus coupled to a QP2010 Ultra mass analyzer and AOC-5000 Plus autosampler, using an HP-5 MS column (30 m × 0.250 mm × 0.25 µm, Agilent) with He as the carrier gas with a flow of 1.55 mL/min, and a split ratio of 20. The temperature gradient started at 60 °C for 2 min, increased at a rate of 10 °C/min until 260 °C and was kept at 260 °C for 2 min. The source and interface temperatures were 260 °C with a solvent cut time of 3 min, a mass range between 50 and 800 *m*/*z*, and a scan speed of 20000. The crude extracts (1 mg/mL) suspended in EtOAc were filtered on a bed of silica-gel, and 1 μL was injected. The identification of compounds was based on the analysis of molecular ions and fragmentary ions by matching of a minimum of eight relevant peaks, comparisons with the literature, and the use of standard compounds (**1**–**5**, **6**, **10**–**14**, **16**, **17**, **19**). 

### 4.5. H NMR Spectroscopic Analysis

Dried crude extracts were dissolved in CDCl_3_ with tetramethylsilane (TMS, 0.05% *v*/*v*) as an internal standard (TMS = 0.00 ppm). The ^1^H NMR spectra were recorded at a frequency of 500 MHz on a spectrometer DRX 500 (Bruker). Each spectrum resulted from 128 scans with pulse widths (PW) of 8.0 μs (30°) and relaxation delays (RD) of 6.0 s.

### 4.6. Multivariate Analysis

Data were processed using Mestre-C (version 4.8.6.0, MestreLab, Santiago de Compostela, Spain), and the FIDs were Fourier transformed with line broadening (LB) = 1.0 Hz. Spectra were referenced to TMS at 0.00 ppm. Spectral intensities (peaks) were integrated into regions of equal width (0.02 ppm) in the range of d 1.40–12.40. Integrated areas were normalized to equal total area and were used in a multivariate analysis. The region δ 7.20–7.30, containing the chloroform peak, was excluded from the analysis. For scaling, the unit variance method was applied to PCA. The ^1^H NMR spectroscopic data of spectra were reduced to 545 integral segments of equal length (0.02 ppm). Principal component analysis was performed with The Unscrambler software (version 10, CAMO Process AS, Norway).

### 4.7. Isolation of Compounds from Piper hemmendorffii

Dried and powdered adult leaves (445 g) of *P. hemmendorffii* were extracted with EtOAc (3 × 1000 mL) yielding a crude extract (27.45 g). Ten grams of extract were suspended in MeOH:H_2_O (250 mL, 85:15) and dechlorophylated in a bed of Celite [62] and extracted with hexanes and then with dichloromethane, yielding 4.51 g and 1.91 g of respective fractions. From the dichloromethane fraction, 1.5 g was submitted to vacuum liquid chromatography [63] using hexane:EtOAc at increasing polarities yielding a sample containing compound **9** (12 mg), which was analyzed by spectroscopic techniques without further purifications [35]. 

### 4.8. Isolation of Compounds from Piper permucronatum

Dried and powdered 6 month old leaves (300 mg) of *P. permucronatum* were extracted using EtOAc (3 × 30 mL) yielding a crude extract (60 mg). The obtained crude extract was subjected to preparative thin layer chromatography (hexane:EtOAc—4:1, three elutions) yielding the major compound **16** (5 mg) identified as sakuranetin. For the isolation of compound **23**, dry seedling leaves (450 mg) of *P. permucronatum* were extracted with EtOAc and yielded 60 mg of crude extract. The crude extract was submitted to silica gel preparative thin layer chromatography (three plates) and eluted with the mixture, hexanes:EtOAc (7:3), yielding 5 mg of **23**.

## Figures and Tables

**Figure 1 plants-10-01085-f001:**
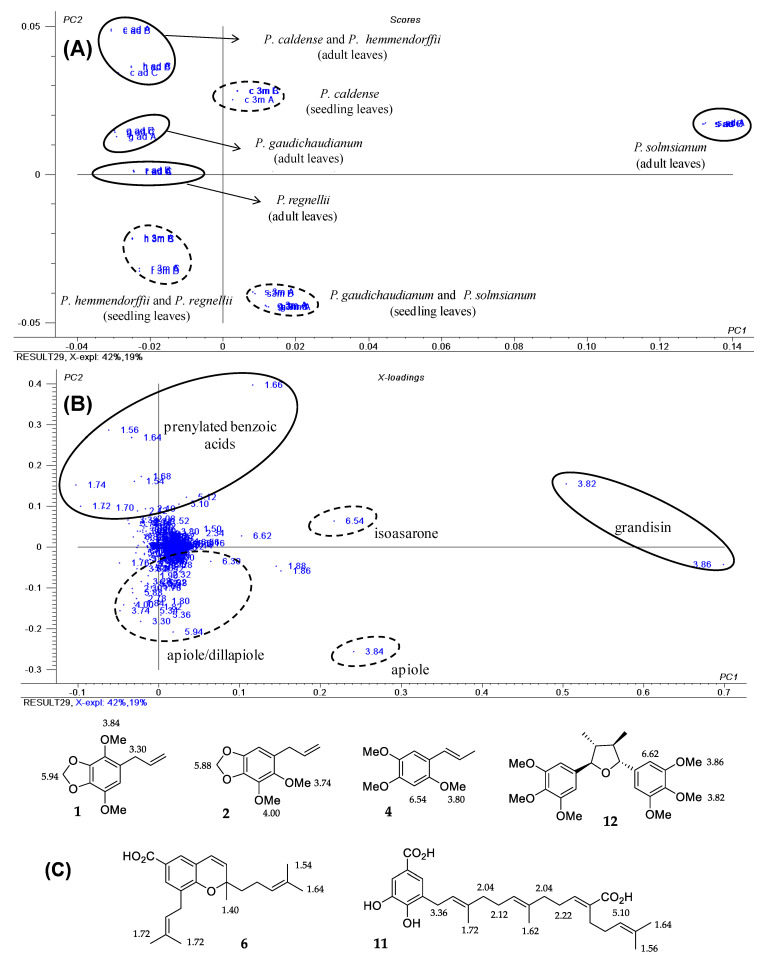
Score plot (**A**) and loading plot (**B**) based on ^1^H NMR data from crude extracts of seedling and adult leaves of *P. gaudichaudianum*, *P. regnellii*, *P. solmsianum*, *P. hemmendorffii* and *P. caldense*. Structures of compounds with corresponding chemical shifts (**C**) of apiol (**1**), dillapiole (**2**) and isoasarone (**4**), grandisin (**12**), gaudichaudianic acid (**6**) and caldensinic acid (**11**), observed in the ^1^H NMR spectra of crude extracts.

**Figure 2 plants-10-01085-f002:**
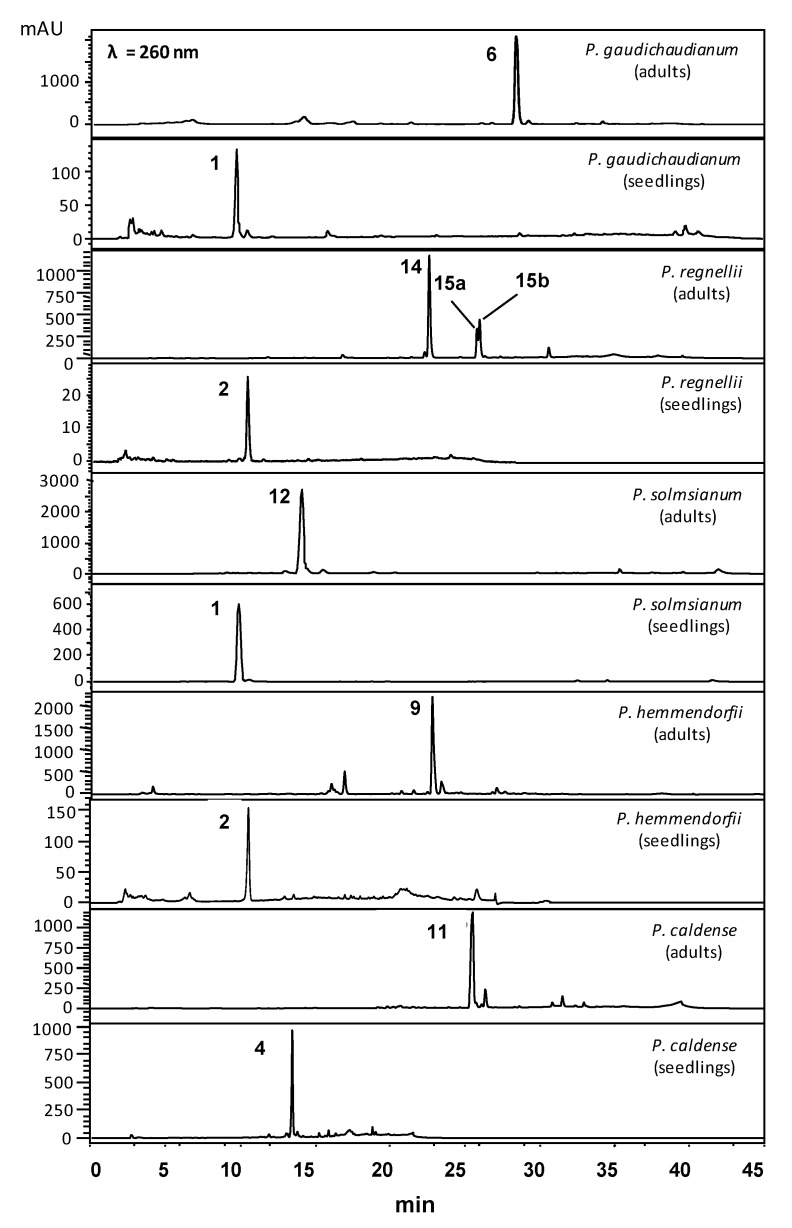
HPLC-UV (260 nm) chromatograms of crude extracts of seedling leaves and adult leaves of *Piper* species (*P. gaudichaudianum*, *P. regnellii*, *P. solmsianum*, *P. hemmendorffii* and *P. caldense*). For the structures of all compounds, see Figure 3 and for ^1^H NMR spectra see Appendix A).

**Figure 3 plants-10-01085-f003:**
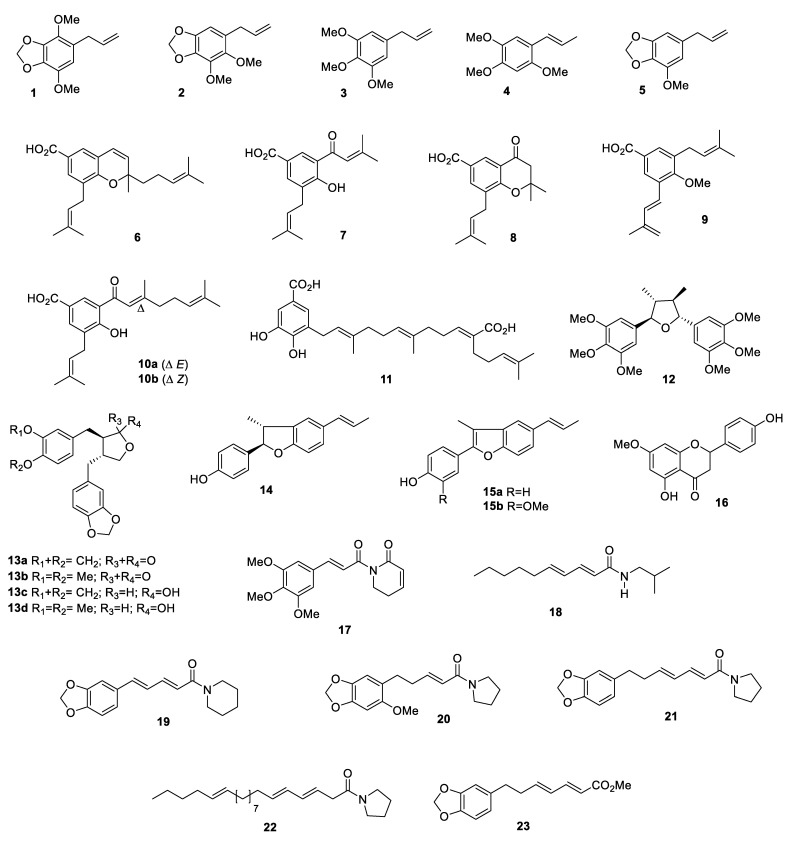
Structures of major secondary compounds identified in adult and seedling leaves of *Piper* species.

**Figure 4 plants-10-01085-f004:**
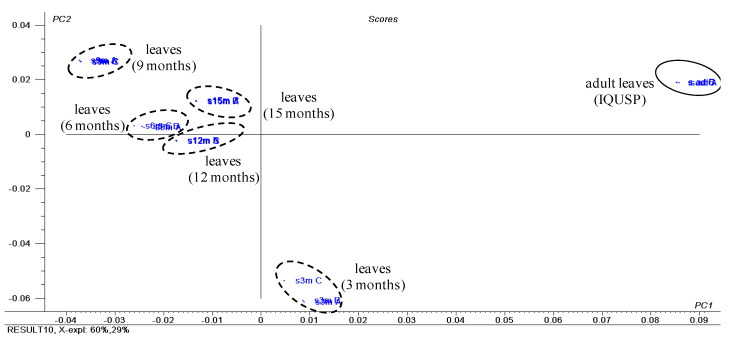
Score plot obtained by PCA using ^1^H NMR data of crude extracts of seedling leaves (seedlings at 3, 6, 9, 12 and 15 months) and adult leaves of *P. solmsianum*. For the loading plot and assignments of chemical shifts of apiol (**1**), myristicin (**5**) and grandisin (**12**) observed in their ^1^H NMR data, see Appendix A.

**Figure 5 plants-10-01085-f005:**
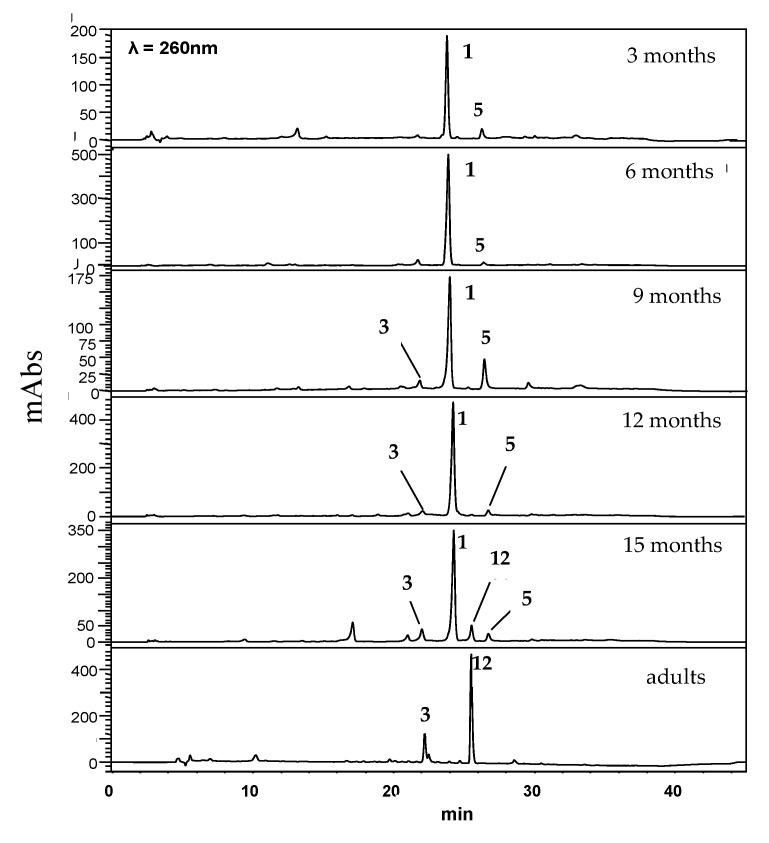
Chromatographic profile (HPLC-UV; λ = 260 nm)) of crude extracts from adult and seedling leaves across the ontogeny of *P. solmsianum.* For structures, see Figure 3.

**Figure 6 plants-10-01085-f006:**
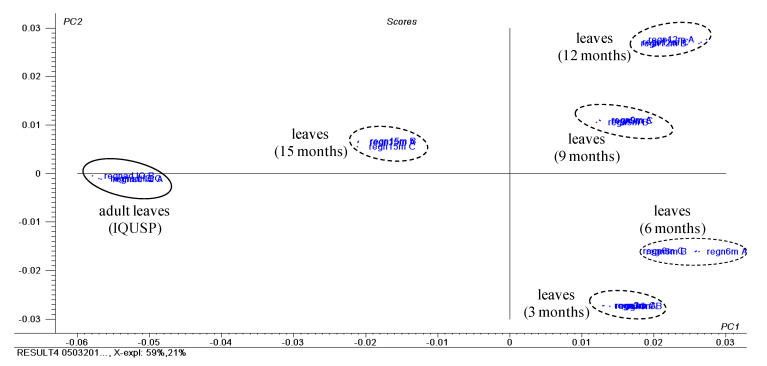
Score plot obtained by PCA using NMR data of crude extracts of seedling leaves (seedlings at 3, 6, 9, 12 and 15 months) and adult leaves of *P. regnellii*. For the loading plot and assignments of chemical shifts observed in their ^1^H NMR data of main compounds, see Appendix A.

**Figure 7 plants-10-01085-f007:**
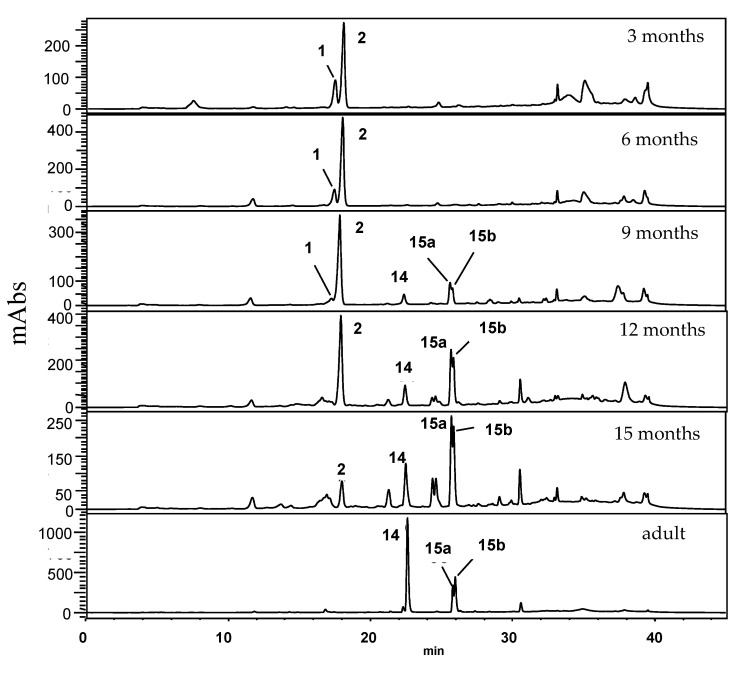
Chromatographic profile (HPLC-UV, λ = 260 nm) of crude extracts from adult leaves and seedling leaves in different developmental stages of *P. regnellii*. For structures, see Figure 3.

**Figure 8 plants-10-01085-f008:**
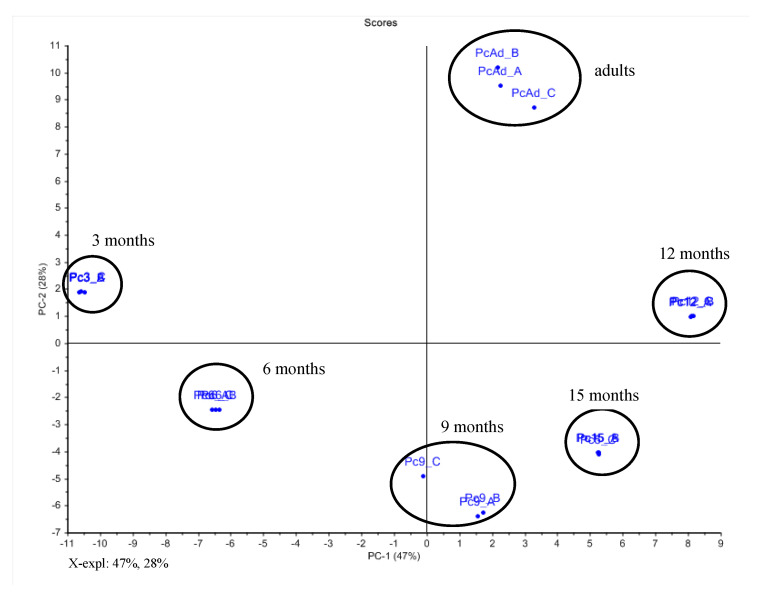
Score plot obtained by PCA using NMR data of crude extracts of seedling leaves (seedlings at 3, 6, 9, 12 and 15 months) and adult leaves of *P. caldense*. For the loading plot and assignments of chemical shifts observed in their ^1^H NMR data of main compounds, see Appendix A.

**Figure 9 plants-10-01085-f009:**
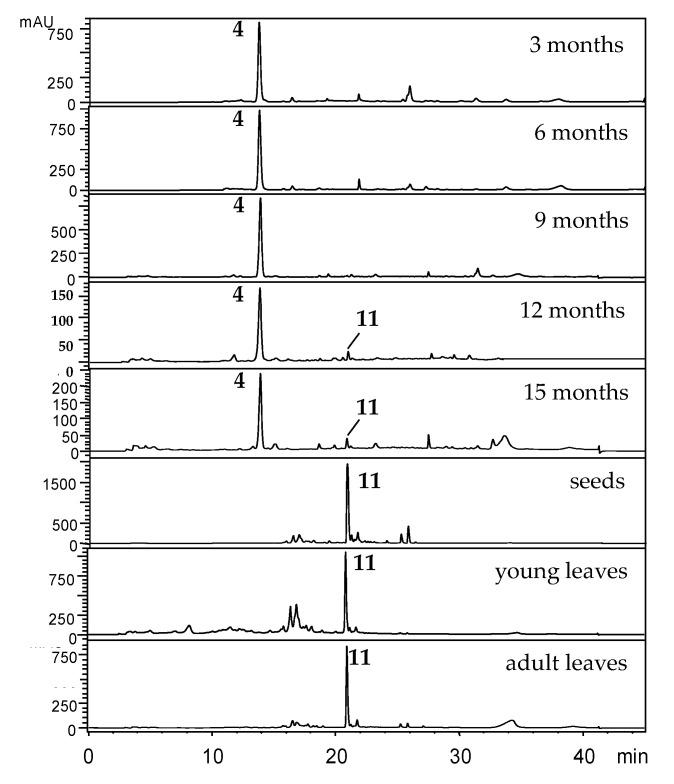
Chromatographic profile (HPLC-UV, λ = 260 nm) of crude extracts from adult leaves and seedling leaves across the ontogeny of *P. caldense*. For structures, see Figure 3.

**Figure 10 plants-10-01085-f010:**
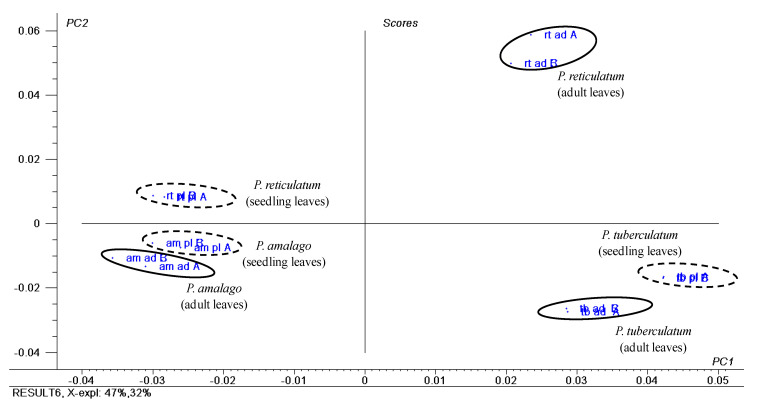
Score plot obtained by PCA of ^1^H NMR data of crude extracts of seedling and adult leaves of *P. tuberculatum, P. amalago* and *P. reticulatum.* For the loading plot and assignments of chemical shifts observed in their ^1^H NMR data of main compounds, see Appendix A.

**Figure 11 plants-10-01085-f011:**
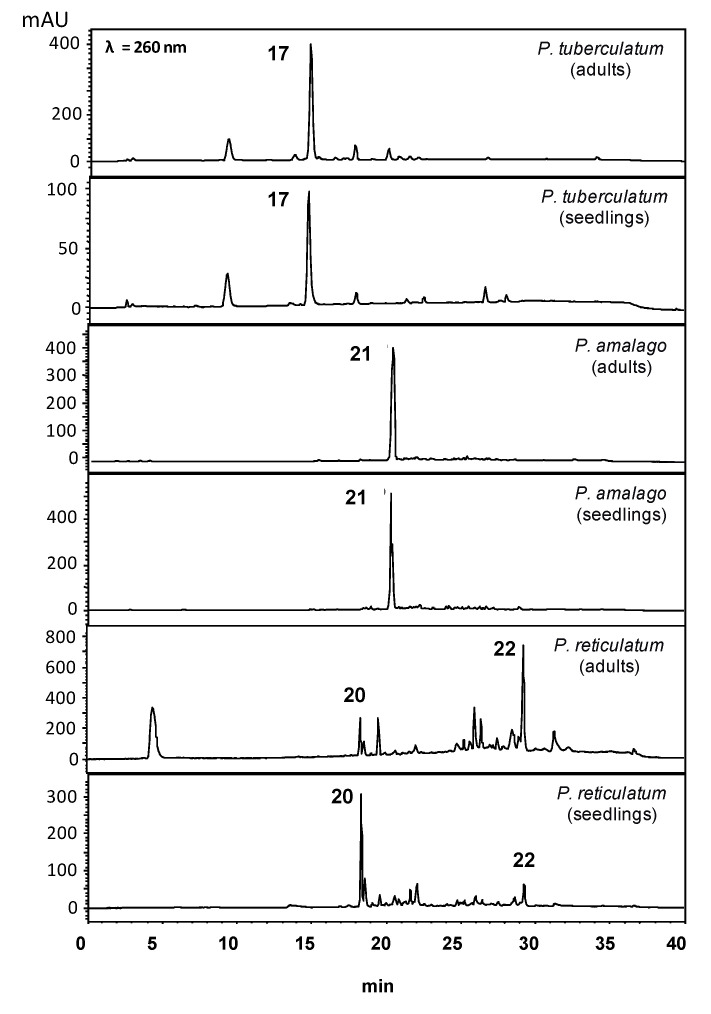
Chromatographic profile (HPLC-UV, λ = 260 nm) of crude extracts of seedling leaves and adult leaves of *Piper* species (*P. tuberculatum*, *P. amalago* and *P. reticulatum*). For structures, see Figure 3.

**Figure 12 plants-10-01085-f012:**
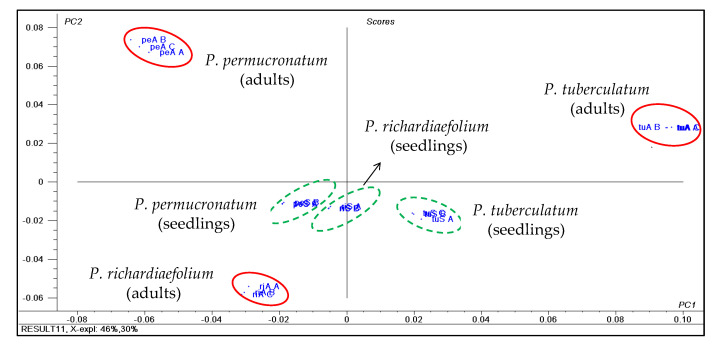
Score plots of the analysis of main components of ^1^H NMR data of the adult and seedling leaves of *P. permucronatum*, *P. richardiaefolium* and *P. tuberculatum*. For the loading plot and assignments of chemical shifts observed in their ^1^H NMR data of main compounds, see Appendix A.

**Figure 13 plants-10-01085-f013:**
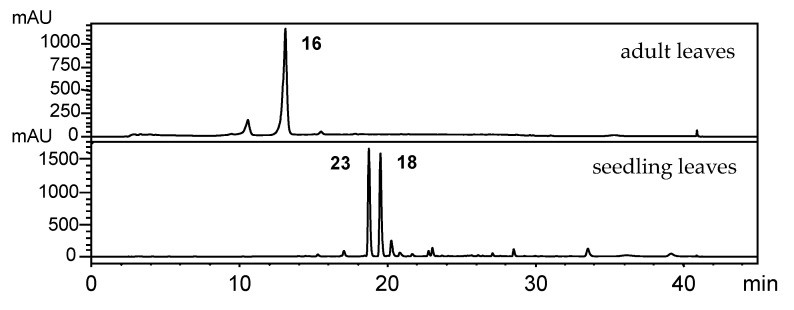
Chromatographic profile (HPLC) of crude extracts from adult leaves and seedling leaves of *P. permucronatum*. Compounds were identified as sakuranetin (**16**), pellitorine (**18**) and the novel compound **23**.

**Figure 14 plants-10-01085-f014:**
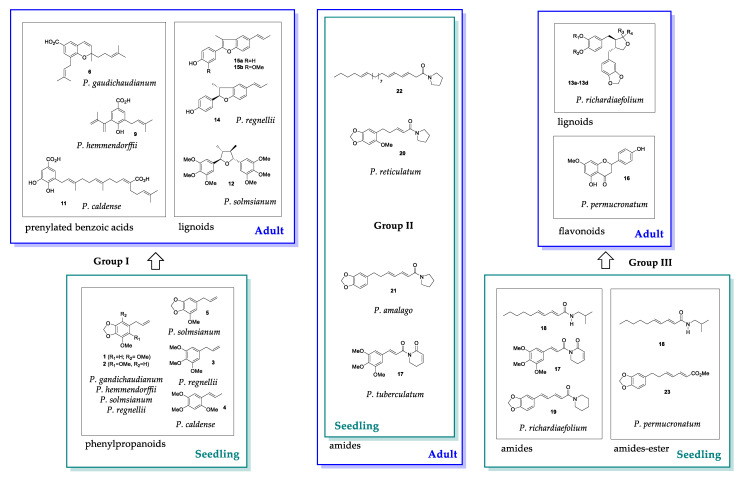
Major ontogenetic changes in secondary compounds of *Piper* species. Group I produces phenylpropanoid in the seedling stage and adults produce benzoic acid derivatives or lignoids; Group II does not change during ontogeny; Group III produces dibenzylbutyrolactones or flavonoids in the adult stage and amides as seedlings.

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
