# Peer review of "Ontogenetic Changes in the Chemical Profiles of Piper Species"

_plants, 2021, doi:10.3390/plants10061085_

Round 1

Reviewer 1 Report

The manuscript describes the phytochemical analysis of seedlings and adult plants of several Piper species 1H NMR spectroscopy combined with multivariate analysis and different chromatographic and spectroscopic methods.

The results showed interesting differences in the chemical composition between seedlings and adult plant and evidenced a regulatory mechanism during ontogeny with a gradual modification on the metabolic profile of Piper plants.

The authors described properly the results. Please, check some minor typing mistakes throughout the text.

Author Response

Thank you for your comments.
We have revised all these typos, mistakes and added additional information.

Reviewer 2 Report

This paper describes the ontological changes in a select group of Piper species, which is a follow-on from a similarly designed study using one Piper species.  The NMR/principal component analysis and LC-MS analytical approach resulted in the author's identifying several groups among the described plant that have either consistent compositions or developmentally variable compositions of phenylpropanoids or alkamides.  The approach taken seems generally valid for categorizing the plants and, overall, the observations, were interesting.

There were some areas that the manuscript could be improved.   Broadly,

1) The discussion is not particularly convincing.  The paragraph starting on line 358 is not well supported.  What evidence is there that the shift in any of the discussed groups of metabolites affect the survival of any of the groups of plants with respect to herbivores?  I don't know what is exactly meant by a "regulatory mechanism during ontogeny" but it is not surprising that secondary metabolism varies with life cycle (or possibly other conditions experienced by the developing plants).  This section of the paper needs much better context, otherwise, I feel that it is overextending the results.

2) The design of the figures is subpar with poorly labeled axes and illegible text on the scores plots.  This reviewer is concerned that the identification of compounds by NMR appears to rely on a few, not particularly diagnostic peaks.  If the eigenvectors from the PCA plot are clear enough to make broad aspects of the compound's identities apparent, it would be helpful to show this in the supplementary material and even for one figure panel in the main text.

3) Figure 7 is very suprising.  It appears to show from the loading pload that the same compound can be a positive or negative loading contributor (and to two PCs).  How is this possible?  This seems to imply that 2-3 different compounds with similar spectral data are being revealed.

4) Figure 8 shows HPLC data for young leaves that seems to directly contradict the first five panels in this figure.  Are the young leaves different from the 3-15 month old leaves?

5) The source of adult leaves is not clear in section 4.1.  There are two mentions of the leaves from the campus and the Institute of Chemistry.  Please clarify this.

6) Line 224: this sentence is confounding.  It says that samples from one species were actually isolated from two species.  What does this mean??

Line 231-33: the data for P. reticulatum seems to be more different, so I don't feel this sentence accurately reflects the data.

The data for compound 16 should be reported in the supplementary data (mentioned on lines 272-77)

The HMBC correlation between H-9 and C-7 is surprising.  How can the authors be sure it is not H-9/-10 to C-8?

Each of the provided NMR spectra should be presented with identical chemical shift ranges to allow proper visual comparison.

Exactly which standards were available and their sources should be made clear in the Methods (as some were not used, noted on lines 233-34).

The article has a significant number of typos, and wording that leaves some sentences unintelligible. 

Specific examples are:

line 16: compounds should be plural

line 21: "...and new long..." should be "...and a new long..."

line 24: the verb "to dribble" usually refers to the bouncing of a basketball.  I have no idea what the intended word is here.

line 32-33:  "Members of the genus have strong interactions with pollination, seed dispersal, and herbivores [4-6]."  This sentence doesn't make sense and should be revised.

end of line 33: "[citations]".  Please add them.

line 49: typo should be "throughout"

line 56-57:  "Such variation is adaptive, as herbivore host searching is limited...".  The meaning of the front end of this sentence is not clear.

line 71: seedlings should be plural

line 112: "synergistic to" should be "synergism with"

line 114-5.  compounds should be singular and the sentence is missing a period.

line 198: methylene hydrogens rather than "methylenic"

line 206: "already" is unnecessary and can be deleted.

line 219-20:  poorly worded and previously is misspelled.

line 234  Piplartine should be capitalized

line 285: extra space

line 315: should be "doublet of doublets"

section 4.2:  ethyl and methanol should not be capitalized.

section 4.4: degree signs should not be underlined

line 460: 15% of methanol or 15% of water? Please clarify

References: there is a mix of abbreviated journal names and full journal names (e.g., ref 19, 20), and a number of references without pages (refs 36, 55) or odd numbers (ref 44).  Refs 46, 47 have formatting issues.

Author Response

This paper describes the ontological changes in a select group of Piper species, which is a follow-on from a similarly designed study using one Piper species.  The NMR/principal component analysis and LC-MS analytical approach resulted in the author's identifying several groups among the described plant that have either consistent compositions or developmentally variable compositions of phenylpropanoids or alkamides.  The approach taken seems generally valid for categorizing the plants and, overall, the observations, were interesting.

There were some areas that the manuscript could be improved.   Broadly,

  • The discussion is not particularly convincing.  The paragraph starting on line 358 is not well supported. What evidence is there that the shift in any of the discussed groups of metabolites affect the survival of any of the groups of plants with respect to herbivores?  I don't know what is exactly meant by a "regulatory mechanism during ontogeny" but it is not surprising that secondary metabolism varies with life cycle (or possibly other conditions experienced by the developing plants).  This section of the paper needs much better context, otherwise, I feel that it is overextending the results.

Response: We have shortened the discussion. The evidenced that we have regarding the potential role of compounds detected in the seedlings are indirect and based on literature data. Therefore, we have added several new references dealing with insecticides or antifungal activities.

2) The design of the figures is subpar with poorly labeled axes and illegible text on the scores plots.  This reviewer is concerned that the identification of compounds by NMR appears to rely on a few, not particularly diagnostic peaks.  If the eigenvectors from the PCA plot are clear enough to make broad aspects of the compound's identities apparent, it would be helpful to show this in the supplementary material and even for one figure panel in the main text.

Response: We have increased the figures to make them legible. Part of the figures (loading and chemical structures with assignment of chemical shifts) were transferred to Supplementary material. The eigenvectors from the PCA plot with a large number of variables became very complex and the vectors are not clear and therefore we decided not to use it.

3) Figure 7 is very suprising.  It appears to show from the loading pload that the same compound can be a positive or negative loading contributor (and to two PCs).  How is this possible?  This seems to imply that 2-3 different compounds with similar spectral data are being revealed.

Response: In fact, the clear assignment is due to the phenylpropanoid isoasarone. The chemical shifts observed in 1H NMR spectra of the crude extracts from seedling of various ages, and some of them can be assigned to caldensinic acid but several are not clearly assignable and we did not assign it as before.

4) Figure 8 shows HPLC data for young leaves that seems to directly contradict the first five panels in this figure.  Are the young leaves different from the 3-15 month old leaves?

Response: The young leaves refers to those from adult plants and that indicated that the age of plants, seedling or adult, are more important in defining the chemical composition.

5) The source of adult leaves is not clear in section 4.1.  There are two mentions of the leaves from the campus and the Institute of Chemistry.  Please clarify this.

Response: We have rewritten this section. Thank you.  

6) Line 224: this sentence is confounding.  It says that samples from one species were actually isolated from two species.  What does this mean??

Response: We have revised the sentence (While in the case of P. reticulatum, the samples were separated from the two species) for “Piper reticulatum seedlings and adults are chemically distinct”.

Line 231-33: the data for P. reticulatum seems to be more different, so I don't feel this sentence accurately reflects the data.

The data for compound 16 should be reported in the supplementary data (mentioned on lines 272-77)

Response: The figure 10 clearly shows the similarities between seedling and adult leaves of P. tuberculatum and P. amalago, but in case of P. reticulatum, the main compound of seedling (20) is still present in the adult, but the minor compound 22 became of the major in adult leaves. We to hope to have clarified in the new text with the sentence: “HPLC-UV data were consistent with results obtained by PCA and revealed that the changes in P. reticularum are more significant between the two stages but with the variable relative content of 20 and 21.”  

The HMBC correlation between H-9 and C-7 is surprising.  How can the authors be sure it is not H-9/-10 to C-8?

Response: In fact, since the signals of H9 and H10 overlap, such correlations could result from both possibilities mentioned, obviously because the chemical shifts of C7 and C8 are very close. Nevertheless, the expansion of the HMBC shows that both correlations are visible and, thus, we added this possibility of correlation in the discussion. In addition, the chemical shifts in the 13C NMR spectrum are very reliable to assign the structure based on similar compounds. Thank you for noticing that.

Each of the provided NMR spectra should be presented with identical chemical shift ranges to allow proper visual comparison.

Response: We replace the expanded version in the supplementary material by the ones with similar chemical shift ranges, but it did not allow accurate visualization as before.

Exactly which standards were available and their sources should be made clear in the Methods (as some were not used, noted on lines 233-34).

Response: In fact, 20 and 21 were not available as standard.

The article has a significant number of typos, and wording that leaves some sentences unintelligible. 

Specific examples are:

line 16: compounds should be plural

line 21: "...and new long..." should be "...and a new long..."

line 24: the verb "to dribble" usually refers to the bouncing of a basketball.  I have no idea what the intended word is here.

line 32-33:  "Members of the genus have strong interactions with pollination, seed dispersal, and herbivores [4-6]."  This sentence doesn't make sense and should be revised.

end of line 33: "[citations]".  Please add them.

line 49: typo should be "throughout"

line 56-57:  "Such variation is adaptive, as herbivore host searching is limited...".  The meaning of the front end of this sentence is not clear.

line 71: seedlings should be plural

line 112: "synergistic to" should be "synergism with"

line 114-5.  compounds should be singular and the sentence is missing a period.

line 198: methylene hydrogens rather than "methylenic"

line 206: "already" is unnecessary and can be deleted.

line 219-20:  poorly worded and previously is misspelled.

line 234  Piplartine should be capitalized

line 285: extra space

line 315: should be "doublet of doublets"

section 4.2:  ethyl and methanol should not be capitalized.

section 4.4: degree signs should not be underlined

line 460: 15% of methanol or 15% of water? Please clarify

Response: We have revised all these typos, mistakes and added additional information.  

References: there is a mix of abbreviated journal names and full journal names (e.g., ref 19, 20), and a number of references without pages (refs 36, 55) or odd numbers (ref 44).  Refs 46, 47 have formatting issues.

Response: We have revised carefully the abbreviations and the format and also added doi for the references when available. The EndNote style available for the journal did not work properly. 

Reviewer 3 Report

The work is interesting, well written and scientifically interesting.
Doubts may arise from too many charts, which are not necessarily transparent (loadings charts). Maybe the classic PCA (scores) dependencies would be enough instead? Alternatively, the cluster analysis could be performed to compare the similarity of the tested samples.
What was the column temperature in the HPLC-DAD analysis?
The above suggestions do not detract from the value of the work, which I find interesting from a practical point of view.
I believe the work may be published on Plants.

Author Response

The work is interesting, well written and scientifically interesting.
Doubts may arise from too many charts, which are not necessarily transparent (loadings charts). Maybe the classic PCA (scores) dependencies would be enough instead?

Response: We have transferred the loading plots and assignments to Supplementary material and left only the Figure 1 with all details as an example.

Alternatively, the cluster analysis could be performed to compare the similarity of the tested samples.

Response: We have tried the cluster analysis for large number of samples from Piper species and we concluded that the results is complex and it is easier to analyze few number of species  (Yamaguchi et al., 2011.  Chemometric Analysis of ESIMS and NMR Data from Piper Species. J Braz Chem Soc 22:2371-U2170).

Response: The column temperature was 40 oC and this is indicated in the experimental.

The above suggestions do not detract from the value of the work, which I find interesting from a practical point of view.
I believe the work may be published on Plants.

Round 2

Reviewer 2 Report

Adjustments to the discussion, and clarifications in the experimental and results section greatly improve the manuscript, and address the concerns of the original review.  Four very small clerical changes are below.

Abstract

Delete "...and of... in this sentence of the abstract.  Adult leaves of P. regnellii accumulate
dihydrobenzofuran neolignans, and of P. solmsianum contain
tetrahydrofuran lignans, and prenylated benzoic acids are found in adult
leaves of P. hemmendorfii and P. caldense. 

Please break the following sentence but replacing the comma with a period.  "Piper gaudichaudianum and P. solmsianum seedlings contain the phenylpropanoid dillapiole, ...."

In the body of the paper

top of Page 5:  "...was the benzoic acid..."  should be "...were benzoic acid..."

line 4, page 5: "...produces..." should be "...produce..."

Author Response

We have revised the minor and critical mistakes. 

The sentence on top of page 5, was change to:

"The main compound in adult leaves of P. hemmendorffii was the benzoic acid (9) [37], and the primary compound in adult P. caldense was caldensinic acid (11) (Supplementary Figures S6 and S7) [38]. "

All the remaining suggestions were accepted.